# Shortage and unequal distribution of infectious disease specialists in Japan: How can we refine the current situation?

Hideharu Hagiya ⬤ *

Department of Infectious Diseases, Okayama University Hospital, Okayama, Japan

* hagiya@okayama-u.ac.jp

## Abstract

### Background

This study aimed to assess the distribution of board-certified infectious disease (ID) specialists at medical schools and Designated Medical Institutions (DMIs) in Japan.

### Methods

Data on the number of board-certified ID specialists was extracted by gender, prefecture, and hospital from the Japanese Association for Infectious Diseases database. The numbers and types of Japanese university hospitals that have a Faculty of Medicine, as well as the DMIs legally determined by the Infectious Diseases Control Law, were collected from the database of the Ministry of Health, Labour, and Welfare of Japan.

### Results

As of November 2022, there were 1,688 board-certified ID specialists in Japan, with 510 employed at 82 university hospitals. Two medical schools had no ID specialists, and six had only one ID specialist. There was no ID specialists in 14.3% of Class I DMIs and 66.7% of Class II DMIs. Additionally, 14.9% of prefectures had no ID specialists at all in their Class II DMIs. The percentage of female doctors among ID specialists was 12.7%, approximately half of the overall male-to-female ratio of medical doctors in Japan.

### Conclusion

The allocation of Japanese ID specialists to medical schools and legally designated healthcare institutes is inadequate and skewed. Female physicians are expected to play a more active role in this increasing demand.

**Data Availability Statement:** The data underlying the results presented in the study are available online but include the individual profiles with their names, which thus should not be disclosed openly. For details, please contact Okayama University Ethics institutional review board committee who

waived the need of informed consent (mae6605@adm.okayama-u.ac.jp).

## Introduction

The global pandemic of coronavirus disease 2019 (COVID-19) has underscored the importance of infectious disease (ID) professionals in maintaining social functions and infrastructure amidst emerging health threats. Board-Certified Physicians of the Japanese Association for Infectious Diseases (board-certified ID specialists) represent the highest qualification for ID doctors in Japan, verifying their expertise in diagnosing and treating various infectious diseases [1]. These specialists are also expected to lead infection prevention and control (IPC) measures in the context of nosocomial infections. However, as of November 2022, there were only 1,688 board-certified ID specialists in Japan [2]. Considering the uncontrolled and uneven distribution of these specialists across hospitals [3], it is evident that ID specialists are not optimally positioned at various healthcare facilities in the country.

Undergraduate education plays a crucial role in imparting fundamental knowledge of clinical infectious diseases to medical students. To enhance ID education, medical schools should employ ID specialists with extensive clinical experience and knowledge in educational positions. However, the allocation of ID specialists in Japanese medical universities remains largely unknown. Additionally, attention should be given to the deployment of ID specialists in Designated Medical Institutions (DMIs) established by law. In Japan, the Act on the Prevention of Infectious Diseases and Medical Care for Patients with Infectious Diseases, also known as the Infectious Diseases Control Law, was enacted in 1998 to combat the spread of highly contagious diseases [4]. The Act classifies DMIs into four categories: (i) DMIs for Specified Infectious Diseases (Specified DMIs), (ii) DMIs for Class I Infectious Diseases (Class I DMIs), (iii) DMIs for Class II Infectious Diseases (Class II DMIs), and (iv) DMIs for Tuberculosis [4]. Class I Infectious Diseases encompass severe conditions such as Ebola hemorrhagic fever, Crimean-Congo hemorrhagic fever, Smallpox, South American hemorrhagic fever, Plague, Marburg virus disease, and Lassa fever. Presently, four medical institutions are designated as Specified DMIs by the Ministry of Health, Labour, and Welfare, while prefectural governors have assigned 56 and 351 hospitals as Class I and Class II DMIs, respectively, to provide care for patients with applicable diseases. The government may address that physicians who have been engaged in the clinical practice of infectious diseases should be employed at such DMIs; however, that is not the case in reality. The employment of ID specialists is not mandated for medical institutions to be designated. In this context, the existing legal framework may not be the most practical approach for effectively managing disease outbreaks.

The objective of this study was to highlight the shortage and imbalanced distribution of board-certified ID specialists in Japan. The findings presented here would have implications for various stakeholders, including governors and administrative officers who are supposed to establish a national IPC platform for future pandemic.

## Methods

Data on the number of board-certified ID specialists by prefecture and hospital were extracted from the database of the Japanese Association for Infectious Diseases [2]. As for the medical schools, we included all national, public, private, and other types of university hospitals (including college hospitals) that have Faculty of Medicine all over Japan, as of November 2022. Those who were registered with affiliated medical institutions outside university hospitals were also considered to belong to university hospitals. The data were evaluated as a whole and stratified by the founding organizations. Data on the number of DMIs were collected from the database of the Ministry of Health, Labour and Welfare of Japan [5]. The deployment status (total numbers and proportions) of the board-certified ID specialists at the Specified, Class I, and Class II DMIs were determined. In addition, the percentage of Class II DMIs with

board-certified ID specialists in the 47 prefectures was determined (the number of Class II DMIs with board-certified ID specialists divided by the total number of Class II DMIs in each prefecture). Deployment was acknowledged if at least one specialist was employed. Data on their sex were not opened to the public; thus, it was provided by the society secretariat for research purposes only in a form of anonymized data. To compare these data with the proportion of female doctors in Japan, the author accessed governmental data, which was open to the public on December 31, 2020 [6]. A need of informed consent was not required by the Okayama University Ethics institutional review board committee because the data were fully anonymized.

Categorical variables are presented as numbers and percentages, and continuous variables are summarized as median and interquartile range (IQR) and analyzed using the Mann-Whitney analysis. The data were analyzed using EZR software, a graphic user interface for R 4.0.3 software (The R Foundation for Statistical Computing, Vienna, Austria) [7]. All reported *p*-values less than 0.05 were considered statistically significant.

## Results

### ID specialists in medical schools (Fig 1)

Data from 82 university hospitals were used, including 42 national, 8 public, 31 private, and 1 other hospital (National Defense Medical College). A total of 510 ID specialists were registered at these hospitals: 234 in national, 42 in public, 229 in private, and five in other university hospitals. There was no ID specialists registered at one national and one private hospital (two universities in total). Only one ID specialist worked at each of six national university hospital. Overall, the median (IQR) number of ID specialists in Japanese university hospitals was 5 (3, 7.75). Significantly more ID specialists were employed at private universities than at national and public universities (N = 50): median (IQR):4 (2, 6.75) vs. 6 (4, 9) [*p* = 0.037].

### ID specialists in DMIs (Fig 2)

Board-certified ID specialists were allocated to all Specified DMIs and 48 Class I DMIs (85.7%) (Fig 2A). Of the 351 Class II DMIs across the country, the ID specialists were registered at 117

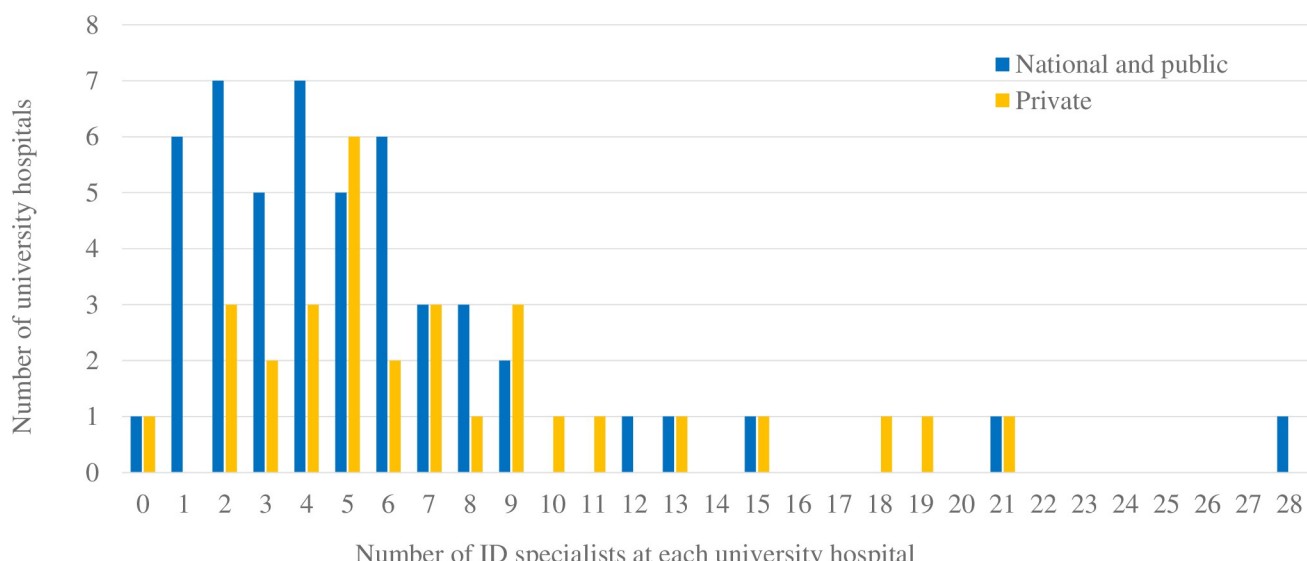

**Fig 1. The number of board-certified infectious disease (ID) specialists in university hospitals in Japan, as of December 2022.**

**(A)**

| Type of DMI | Total number of hospitals | Total number of allocated beds | Number (proportions) of DMI with ID specialist |
|---|---|---|---|
| DMI for Specified Infectious Diseases | 4 | 10 | 4 (100%) |
| DMI for Class I Infectious Diseases | 56 | 112 | 48 (85.7%) |
| DMI for Class II Infectious Diseases | 351 | 1,766 | 117 (33.3%) |

**(B)**

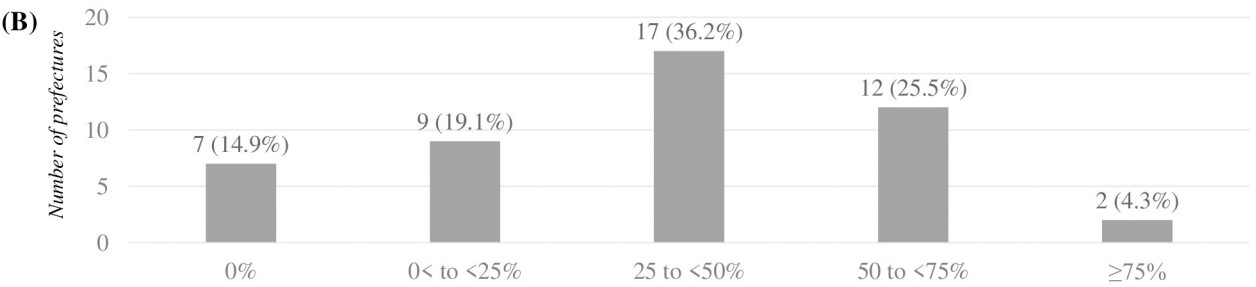

*Deployment sufficiency proportions of board-certified ID specialists at Class II-DMIs in each of 47 prefectures in Japan*

**Fig 2.** Deployment of board-certified infectious diseases (ID) specialists at Designated Medical Institutions (DMIs) for Specified, Class I, and Class II Infectious Diseases (A) and the percentage of Class II DMIs hiring such specialists across 47 prefectures in Japan (B), as of October 2022.

(33.3%) institutes. There were no board-certified specialists in Class II DMIs in seven prefectures (14.9%) (**Fig 2B**). The employment rates of 33 prefectures were less than 50%, which is equivalent to around 70% of all the prefectures.

## Female ID specialists (Fig 3)

Of the 1,688 board-certified ID specialists in Japan, 215 (12.7%) were female. The highest proportion of female ID specialists was observed in Ibaraki (37.5%), followed by Fukushima (21.4%), and Tokushima (21.4%). There were no female ID specialists in 12 of the 47 prefectures (25.5%). The overall percentage of female doctors in Japan was 22.8%; by age, 36.3% were in their twenties, 31.2% in their thirties, 28.3% in their forties, 18.8% in their fifties, 11.7% in their sixties, and 9.5% in their seventies or older. These facts suggest that although the proportion of female doctors has recently increased across the country, the number of those who chose ID as their specialty remains low.

## Discussion

The present study has revealed a skewed distribution of ID specialists in Japan. Firstly, the number of ID specialists at each university hospital varies significantly among academic institutions, ranging from zero (2.4%), one (7.3%), or two (12.2%) to more than ten (14.6%). While the appropriate number of ID-specialized faculty members in medical schools remains uncertain, having zero or one is insufficient, considering their expected contributions to undergraduate education and medical services. The author believe that even two or three full-time ID specialists would be inadequate to meet the increasing demands in this era of emerging infectious diseases and antimicrobial resistance. Secondly, ID specialists are not adequately deployed in legally determined DMIs. Of note, 14.3% (8 of 56) of Class I DMIs did not employ any ID specialists, and no Class II DMIs had any ID specialists among 14.9% (7 out of 47) of the prefectures. Thus, this legal framework may not always function as intended, and

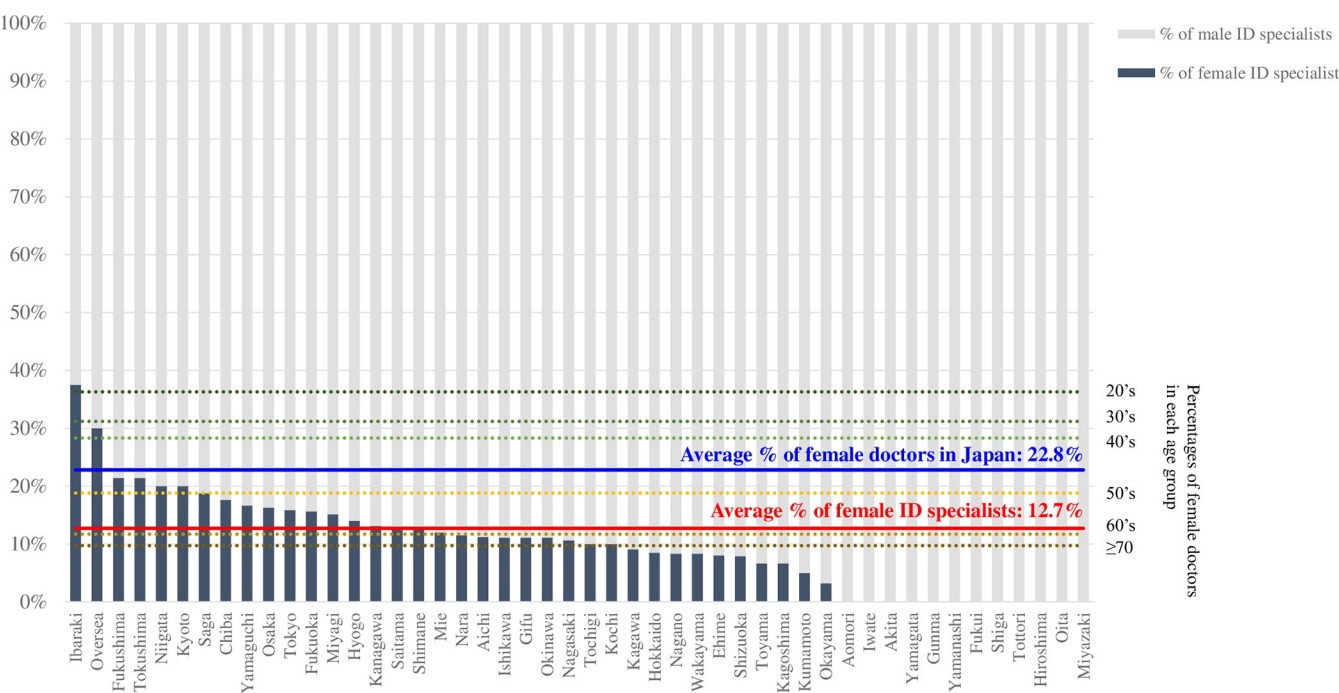

**Fig 3. Proportions of female board-certified infectious disease (ID) specialists in Japan, by prefecture.** Data on ID specialists were extracted from the open data source of the Japanese Association for Infectious Diseases on December 1, 2022. In comparison, the proportion of general female doctors in Japan was given by age as of December 31, 2020.

multifactorial efforts are necessary to strengthen the social role of DMIs. The author has also identified a gender imbalance among Japanese ID specialists, with the proportion of female ID specialists accounting for only approximately half of the total proportion of female doctors in Japan (12.7% vs. 22.8%). These findings, along with the following discussion, are crucial for considering the future of ID specialists in this country.

To adequately prepare for and address future emerging threats posed by infectious diseases, a greater number of ID specialists are required. As of November 2022, prevalence of ID specialists in Japan was 1.35 per 100,000 population, which was approximately three-times higher than in Korea (0.47 in 2019) [8]. Although clear data for other developed countries is unavailable, the situation appears to vary in each country. For example, the results of the 2023 ID fellowship match in the United Staes were disastrous, with 44% of ID fellowship programs remaining unfilled [9], indicating a potential future decline in the number of ID specialists. While, there has been an increasing interest in ID specialization among medical students in France recently [10], which may contribute to a rise in the number of ID specialists.

First and foremost, undergraduate education plays a crucial role. To gain comprehensive knowledge in the field of clinical infectious diseases, high-quality educational opportunities based on a systematic curriculum are essential. ID-specialized faculty members are expected to play a central role in this regard. However, as demonstrated, there is a shortage of such human resources in many Japanese medical schools. The present work lacks data on the educational positions of each ID specialist employed at universities; some may not have any educational responsibilities. Consequently, the number of ID specialists who were faculty educators in each medical school might be overestimated in this study. With fewer educational opportunities guided by ID specialists, fewer undergraduate students would develop an interest in clinical infectious diseases. Although the reason for the higher number of ID specialists in private

universities remains unclear, our results indicate that this issue should be particularly discussed and addressed at national universities.

Secondly, the establishment of an ID training system for resident doctors is crucial. In Japan, there is a postgraduate clinical training system that requires residents to participate in a mandatory two-year course program. During this period, residents are expected to gain experience with numerous clinical cases of infectious diseases. The presence of ID specialists would greatly enhance their learning experience and increase the likelihood of them specializing in this underrepresented field in the future. Furthermore, the author believes that other training opportunities for individuals with diverse backgrounds (e.g., disabled or doctors on parental leave) should also be available to encourage a wide range of trainee candidates.

Securing ID specialists at DMIs is a matter of utmost importance. In this regard, financial assistance from prefectures or municipalities would greatly benefit the designated institutions. The author contends that enhancing the capabilities of DMIs will positively impact the public health of local communities. Additional strategies for enhancing the functionality of DMIs include establishing a national certification system for medical doctors and implementing government-led deployment of qualified physicians. While the current legal framework does not mandate DMIs to employ ID specialists, the author suggests that educational initiatives and social factors must be taken into account to enable DMIs to have full-time ID specialists.

To address these urgent issues, the author anticipates active participation of female physicians in the field of infectious diseases. Gender differences among doctors have been discussed in literature from various perspectives. Notably, several studies have demonstrated that female doctors outperform their male counterparts in medical procedures such as central venous catheterization [11], operations [12], and cardiopulmonary resuscitation [13]. Nevertheless, female doctors are often provided with fewer opportunities [14] and have fewer positions of responsibility [15]. These facts exemplify significant gender inequality. In addition to this concerning situation, the clinical and academic careers of female doctors are vulnerable to various external factors. For example, during the COVID-19 pandemic, the percentage of manuscript submissions by female scientists decreased to a greater extent than that by male scientists, possibly due to the increased burden of childcare as a result of the closure of daycare centers and schools [16, 17]. This trend was, however, necessarily not true in Japan, where publications on COVID-19 by female researchers reportedly increased by 23.7% amid the pandemic [18].

To address the unpopularity of ID specialties among female doctors, it is essential to highlight their presence and acknowledge their diversity. A certain portion of ID specialists work as consultants and do not engage in the duties of the physician in charge. In addition, some of them have no night duties or emergency calls; thus, it is comparatively possible to balance childcare and work as full-time doctors. Therefore, female doctors or female students with childcare responsibilities may benefit from choosing ID as their future specialty. Hopefully, increased and active participation of female physicians in clinical infectious diseases will help alleviate the problem of insufficient ID specialists.

In Japan, there are four major medical societies associated with infectious diseases: the Japanese Association for Infectious Diseases, the Japanese Society of Chemotherapy, the Japanese Society for Clinical Microbiology, and the Japanese Society for Infection Prevention and Control. As of December 1, 2022, four of four presidents (100%), three of four vice presidents (75%), and 65 of 75 board directors (86.7%) of these societies are men [19–22]. In order to incorporate women's perspectives into the establishment of a comfortable working environment for female physicians and to provide female role models, it is recommended to appoint additional female ID specialists to such leadership positions.

In conclusion, this study highlights the skewed allocation of ID specialists in Japan, particularly the shortage in medical schools and legally designated healthcare institutions. The

development of comprehensive education and training curricula at the undergraduate and postgraduate levels is crucial, especially with regards to increasing female representation in the field. The global COVID-19 pandemic has emphasized the essential role of ID specialists in mitigating health threats. Currently, there is a pressing need to address the shortage and disproportionate distribution of ID specialists in Japan.

## Acknowledgments

The author utilized ChatGPT for English proofing, while no single sentence or idea was generated by the open artificial intelligence.

## Author Contributions

**Conceptualization:** Hideharu Hagiya.

**Data curation:** Hideharu Hagiya.

**Formal analysis:** Hideharu Hagiya.

**Investigation:** Hideharu Hagiya.

**Methodology:** Hideharu Hagiya.

**Visualization:** Hideharu Hagiya.

**Writing – original draft:** Hideharu Hagiya.

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
