## [Decision Letter · Decision Letter 0]

21 Jun 2023

PONE-D-23-04516Biased allocation of infectious diseases specialists in JapanPLOS ONE

Dear Dr. Hideharu Hagiya

Thank you for submitting your manuscript to PLOS ONE. After careful consideration, we feel that it has merit but does not fully meet PLOS ONE’s publication criteria as it currently stands. Therefore, we invite you to submit a revised version of the manuscript that addresses the points raised during the review process.

After the initial examination, it appears that your manuscript will need several revisions. These are not meant to dismiss your work but rather to enhance its impact and strengthen its scientific rigor. We want to ensure your manuscript receives the utmost level of evaluation from experienced reviewers in the field, to provide valuable insights and recommendations. However, I regret to inform you that we have faced considerable challenges in finding suitable reviewers. Despite reaching out to many potential reviewers, we have not been successful thus far. Please accept our sincere apologies for this unusual delay in the reviewing process.

We understand that this delay may affect your publication timeline. We truly appreciate your patience and understanding in this matter. We are doing our utmost to expedite the review process and are continuously reaching out to additional experts in the field.

Thank you once again for choosing our journal as a platform to share your valuable research. Please rest assured that we remain committed to handling your manuscript as swiftly and efficiently as possible.

We look forward to receiving your revised manuscript.

Kind regards,

Takashi Watari

Academic Editor

PLOS ONE

Journal Requirements:

None to declare 

None to declare

Reviewers' comments:

Reviewer's Responses to Questions

**Comments to the Author**

1. Is the manuscript technically sound, and do the data support the conclusions?

Reviewer #1: Yes

2. Has the statistical analysis been performed appropriately and rigorously? 

Reviewer #1: Yes

3. Have the authors made all data underlying the findings in their manuscript fully available?

Reviewer #1: Yes

4. Is the manuscript presented in an intelligible fashion and written in standard English?

Reviewer #1: Yes

5. Review Comments to the Author

Reviewer #1: General comments

This article discusses the uneven distribution of infectious disease physicians in Japan. This is an issue that has not been discussed much until now, but with the COVID-19 pandemic, it has become an issue that should be seriously considered. This is an important paper that provides important suggestions for medical policy.

However, it is also true that many issues remain that need to be corrected, and these should be addressed through the following comments.

Major comments

1. The author's opinion is stated from the Introduction, and in some places the discussion is based on facts that are not with appropriate evidence. This point needs to be addressed.

2. I agree that it is important to increase the number of female infectious disease physicians in the future, as it is stated that there are few women among infectious disease physicians. However, there is a problem with the reason that housework and being an infectious disease physician can be compatible. This is because I think it could be misinterpreted that the author thinks it is common for women to do household tasks.

3. It is stated that the gender and age of the infectious disease physicians were obtained from the Japanese Association for Infectious Diseases. If so, I think that the authors need to provide the reason why they were able to receive that data from the Japanese Society of Infectious Disease. The authors have stated that they have not undergone ethical review in conducting their research, but I think there is a possibility that they handled personal information. If the authors received post-processed data from the Japanese Society for Infectious Diseases, there would be no issue.

4. The small number of infectious disease physicians in Japan has been raised as a problem, and information on how the situation is in other developed countries is considered important. Europe, the U.S., and Australia seem to have more infectious disease physicians per 100,000 people than Japan, while France, for example, does not. Please incorporate the perspective of how it is in other countries.

Minor comments

Line 59 I think it would be better to state somewhere what is included in Class I infectious diseases.

Line 65 The government requires that "physicians who have been engaged in the practice of infectious diseases" be engaged in Class I DMIs. The problem is that the phrase "engaged in infectious disease practice" is sometimes interpreted expansively and is deceptively used by physicians who have no training in infectious disease practice. Please revise for clarity on this point.

Line 66 It is unusual for the author's opinion to be expressed in the Introduction section. However, the author's concern is valid, so please revise the wording.

Line 94 When writing a paper using the EZR software, there are references that should be cited. Please confirm.

Line 118 In the discussion of the gender ratio of infectious disease specialists, there is a category of "overseas," which I think should be cut.

Line 164 Is there any evidence that the situation will improve if the government controls the number of infectious disease specialists? Please add this taking into account the outline of the new specialist system.

Line 171 I think it is unlikely that the fact that female physicians reportedly outperformed male physicians in skill is important to the discussion of this paper.

Line 175 It is true that the percentage of female among infectious disease specialists is less than that of all physicians, but it is more in dermatology and obstetrics and gynecology. I think it is unclear whether the low percentage of female among infectious disease physicians is really a result of gender inequality.

Line 177 The number of female authors in medical journals in Japan does not necessarily seem to be decreasing based on a article in 2021 ( J Med Internet Res. 2021 Apr 12;23(4):e25379. Please revise the content to take into account the latest data.

Line 183 Not all infectious disease physicians are consultants. Infectious disease physicians in large medical offices that have produced many infectious disease physicians work both night shifts and as attending physicians. Furthermore, some infectious disease specialists are affiliated with departments other than the Department of Infectious Diseases. Please revise the wording.

Line 189 It is not clear what role the large number of men as representatives and councilors of infectious disease societies plays in drawing conclusions from the results obtained in this study.

6. PLOS authors have the option to publish the peer review history of their article (what does this mean?). If published, this will include your full peer review and any attached files.

Reviewer #1: No

---

## [Author Response · Author response to Decision Letter 0]

14 Aug 2023

We are grateful to both the Editor and Referees for their constructive critique and recommendations to improve our manuscript. We have made every effort to address the issues raised and have responded to all comments. The manuscript has been rechecked and the necessary changes have been made in accordance with the referees’ suggestions. The changes to the revised manuscript have been indicated with red font. I look forward to working with you and the referees to move this manuscript closer to publication. Please find our point-by-point responses to the Editor’s and Referee’s comments below.

Note:

#1 The title has been changed to address more clearly what the author state in this manuscript.

#2 Author’s affiliation has been changed because it was changed during the submission process.

#3 Overall, English grammar have been checked again and revised as appropriate by using ChatGPT.

Thank you for your consideration. I look forward to hearing from you.

---

## [Editor Report · Decision Letter 1]

4 Sep 2023

Shortage and unequal distribution of infectious disease specialists in Japan: How can we refine the current situation?

PONE-D-23-04516R1

Dear Dr. Hagiya,

We’re pleased to inform you that your manuscript has been judged scientifically suitable for publication and will be formally accepted for publication once it meets all outstanding technical requirements.

Kind regards,

Takashi Watari

Academic Editor

PLOS ONE

Additional Editor Comments (optional):

I believe this paper offers intriguing insights into the systemic challenges that Japan faces after the COVID-19 pandemic. While there may be limitations, I consider it a worthy contribution for publication.
---

## [Editor Report · Acceptance letter]

12 Oct 2023

PONE-D-23-04516R1 

Shortage and unequal distribution of infectious disease specialists in Japan: How can we refine the current situation? 

Dear Dr. Hagiya:

I'm pleased to inform you that your manuscript has been deemed suitable for publication in PLOS ONE. Congratulations! Your manuscript is now with our production department. 

Kind regards, 

on behalf of

Dr. Takashi Watari 

Academic Editor

PLOS ONE